# Use of Mobile Phones in Teaching English in Bangladesh: A Systematic Review (2010–2020)

Prodhan Mahbub Ibna Seraj [1] , Blanka Klimova [2,]* and Hadina Habil [3]

1   School of Education, Universiti Teknologi Malaysia, Johor 81310, Malaysia; mahbub@graduate.utm.my
2   Department of Applied Linguistics, Faculty of Informatics and Management, University of Hradec Kralove, 500 03 Hradec Králové, Czech Republic
3   Language Academy, Universiti Teknologi Malaysia, Johor 81310, Malaysia; hadina@utm.my
*   Correspondence: blanka.klimova@uhk.cz

**Abstract:** The use of mobile devices for English language teaching (ELT) is increasing rapidly all over the world. This review study surveys the empirical research on using mobile phones in ELT published in Scopus and Web of Science indexing journals from 2010 to 2020 in Bangladesh. Out of 103 studies, 11 studies met the criteria of this study to analyze the effects of mobile phones on ELT. The findings show that the major research trends of these studies aim at teachers' professional development using mobile phones for teaching language. The findings also reveal that the use of mobile phones is effective in ELT through facilitating feasible, ubiquitous, and effective learning environments with some limitations, i.e., an issue with charging, a small screen, affection, and a lack of teacher confidence. Of the studies conducted, 83% have employed a qualitative research design for investigating learners' readiness and concepts on the use of this device. In addition, there is a lack of empirical studies with the intention to observe and justify the effect of mobile phones on developing learners' language skills. There is also a lack of evidence describing which mobile applications are effective for developing relevant language skills. Overall, the results of this systematic review might be applicable in the context of similarly developing countries, as well as triggering empirical research in the field of technology-enhanced ELT in these countries.

**Keywords:** mobile phone; English language teaching; Bangladesh; benefits; limitations

## 1. Introduction

The mobile phone is interchangeably used with a smartphone, cell phone, personal digital assistant (PDA), or wireless handheld device (WHD) [1]. Due to its portability, ubiquity, comfortability, versatility, social networking, context-sensitivity, uniqueness, and ease of accessibility, this device has become the most widely used handheld device on the globe for performing multiple functions [2]. With the numerous applications available on a mobile phone, its use is not limited to communication; instead, it pervades all facets of human life [3].

On the other hand, many English as a foreign language (EFL) context continue to use conventional language teaching and learning methods. The conventional lecture-based and teacher-centered teaching approach is ineffective for language teaching and learning in EFL contexts [4]. Thus, traditional methods must inevitably be on par with the advent of technology to complement English teaching in EFL contexts [5].

In some EFL contexts, the mobile application achieved its aims in teaching English as a foreign language [6]. For developing learners' English language skills, mobile phones in an English language learning course were effective in Malaysia [7], Saudi Arabia [8], South Korea [9], Czech Republic [10], Russia [11], or Indonesia [12,13]. The use of mobile phones was effective for developing learners' oral communication skills [7] and vocabulary [14] at the tertiary level. There was a strong positive relationship between the use of mobile phones and learners' language performance [8,9]. However, the use of mobile

phones was significantly effective for developing young [13] and secondary [12] learners' speaking skills, tertiary level learners' listening skills [11], and vocabulary [10]. Thus, mobile phones have been used to develop English language skills for learners at different educational levels.

Numerous review studies were conducted on different occasions, and the findings showed both the relevance and efficacy of using mobile phones in the field of education. Additionally, these review studies offered a summary of the various issues associated with using a mobile phone to teach and learn language skills in various contexts. Between 2000 and 2020, researchers discovered that several studies had examined the use of various mobile phone applications for improving learners' different language skills, i.e., listening, writing, speaking, and reading [15–19]. Additionally, some review studies identified 21 [15] and 16 [19] EFL contexts of the studies on mobile phones in 2001 and 2015.

In Bangladesh, mobile devices for English language teaching were launched by a project, 'English in Action'—in 2008. This project targeted developing English language skills of 25 million primary and secondary students and adult learners using mobile-phone-based materials. The project was run with the help of BBC World Service Trust and BBC Learning English, and the program entitled "BBC Janala"—an initiative that provides English language lessons to learners via their mobile phones as part of the more comprehensive English in Action program in Bangladesh [20,21]. Moreover, at present, there are 165.615 million active mobile phone users and 99,984 million active internet users [22] in Bangladesh, which has a population of approximately 168.1 million [22]. Additionally, the government of Bangladesh has launched the Vision 2021–2041, which includes a master plan (2012–2021) for the use of information communication technology (ICT) at all levels of education, with the goal of increasing access to education for all, improving the standard of education, producing skilled manpower, and eradicating digital divide/discrimination in order to keep pace with the fourth industrial revolution. To accomplish these goals, the master plan's objectives are to improve the teaching and learning environment, teachers' professional and ICT skills, and to standardize teaching and learning materials, as well as to develop qualified human resources that meet current needs [22]. As a result, policymakers, teachers, and educators in Bangladesh attempt to incorporate technology, especially mobile phones, into the teaching of English language skills to develop students' language skills.

However, there is a lack of evidence in review studies for observing and judging the present status in the light of past experience and knowledge of using mobile phones in English language teaching (ELT) in Bangladesh. Therefore, this study investigates the research trends and effectiveness of using mobile phones in ELT in Bangladesh from 2010 to 2021.

## 2. Literature Review

Different research studies have been conducted in order to examine the effectiveness of mobile use in ELT for different levels of students, i.e., from young to adult learners in different English as a foreign language (EFL) context. The research trends of some studies reveal that mobile-based learning is more useful and effective than conventional or traditional teaching methods [5,23]. Some other studies point out the challenges and benefits of using mobile devices in ELT, i.e., [24–26]. The research trends of other studies explore teachers' and learners' attitudes and beliefs regarding using mobile phones [27], as well as teaching techniques implemented thanks to mobile phones [28]. The findings indicate that for developing learners' reading [29] and listening skills, the use of mobile phones is effective in EFL contexts such as Malaysia [30]. In addition, students can enrich their vocabulary through the use of mobile phones [31,32]. Moreover, the use of mobiles is effective for developing pronunciation [33,34], which enhances EFL learners' speaking skills [13].

Similarly, the use of mobile phones is also effective for all levels of learners, i.e., from young to adult. Kindergarten learners acquire more benefits when learning the English

language through mobile phones rather than traditional education tools [35]. In addition, the use of mobile phones has a significant impact on developing the writing skills of college students [9]. At a tertiary level, educational mobile phones work as a tool for developing EFL learners' speaking skills [7] and vocabulary [32]. Moreover, the use of mobile phones is very effective for facilitating learning environments through sharing and communication among learners [26,27]. Thus, the previous studies in different EFL contexts reveal that that mobile phone is an effective tool for developing learners' language skills, regardless of the level of learners' education.

Furthermore, several review studies, i.e., [16,29,36–38] on the use of mobile phones in ELT, confirm the information above. The study exploring the characteristics and research trends of using mobile phones concludes that the most of the studies dealt with teaching and learning vocabulary in ELT. The findings also suggest that research in the field of using mobile devices accelerated in the years following 2008 and peaked in 2012. Numerous studies conducted their study without reference to any theoretical context. The field was dominated by applied and design-based research, with the majority of these studies using quantitative research methods. The review study [37] describing the period between 2009 to 2018 on the use of mobile phones in ELT focused on the pedagogical approaches of using mobile phones for learning language skills. This study shows that research in the field of ELT was increasing with the passage of time. The findings also indicate that the most papers were published in 2017, while the fewest were published in 2014. English was the most frequently chosen target language, and the most emphasis was placed on writing, speaking, and vocabulary. Another review study [38], depicting the period between 2010 and 2016, illustrates that most of the studies targeted tertiary level learners for using mobile devices. Among the other significant results of this study is that the majority of research examined the effect of mobile learning on student achievement. Furthermore, the most frequently studied subject domain was language instruction. The findings of the most recent review study [29], dating from 1 January to 30 September 2020, focused on the use of mobile phones for developing learners' reading skills. The findings of this study indicate that mobile learning is becoming a more prominent aspect of education, owing to the fact that it provides an excellent opportunity for foreign language learning. Its primary benefits include increasing the learner's cognitive ability, the learner's willingness to study in both formal and informal environments, increasing the learner's autonomy and trust, and promoting customized learning by assisting low-achieving students in achieving their study goals.

Overall, the results of the review studies described above illustrate that the use of mobile phones in ELT is increasing rapidly in ELT. This technology is being used for developing learners' listening, reading, writing and speaking skills. The published research findings show that this technology is effective for all levels of learners, i.e., from young to adult. Thus, the findings of the review demonstrate that the use of mobile phones is effective in developing learners' language skills in different contexts.

Nevertheless, the findings of this literature review confirm that there is still a lack of evidence on the use of mobile phones in ELT in Bangladesh. Therefore, the purpose of this study is to highlight the research trends and the effectiveness of the use of mobile phones in ELT in Bangladesh.

## 3. Methods

The methodology of this systematic review consists of three process, i.e., search, selection and data analysis processes. The principles of Preferred Reporting Items for Systematic Reviews and Meta-Analyses (PRISMA) were followed to present the results of the searching and selection process of this systematic literature review (Figure 1) [39]. PRISMA is an evidence-based set of items envisioned to prepare authors to report on a wide range of systematic reviews and meta-analyses. PRISMA has been successfully used in many earlier educational studies [39,40]. The process of this study is described as follows:

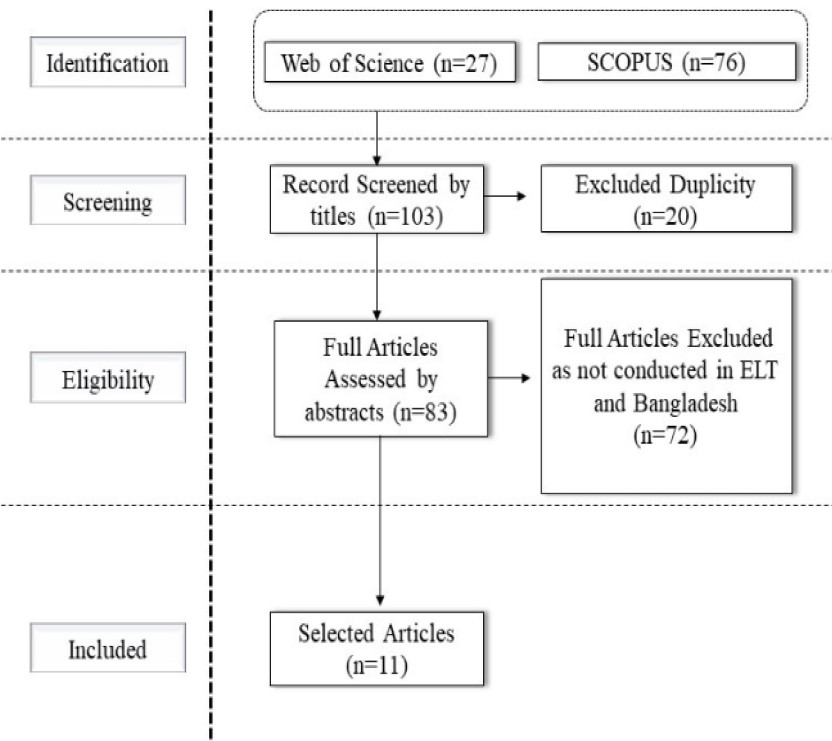

**Figure 1.** PRISMA flow chart.

### 3.1. Search Process

The authors carried out a literature search on the topic of mobile phones in ELT in Bangladesh in two databases that are Scopus and Web of Science (WoS), which have the greatest coverage and scope. The search period was executed for studies published between 1 January 2010 and 31 December 2020 on this topic. The search-chunked keywords were as follows: mobile phones AND language teaching and learning, mobile technologies AND language teaching and learning, mobile applications AND English language, mobile devices AND language teaching and learning, smartphones AND language teaching and learning. The keywords were combined and used to conduct database and journal searches. The terms were searched using AND to combine the mentioned keywords and OR to exclude potential search duplication. Additionally, a backward search was performed, in which the references of retrieved articles were screened for potentially relevant articles that the authors' searches may have missed.

### 3.2. Selection Process

The selection process based on the criteria mentioned in the PRISMA model (Figure 1) was conducted in four phases, i.e., identification, screening, eligibility, and inclusion. Hence, the number of articles in both databases (WoS and Scopus) was 103. The authors detected 27 papers in Web of Science and 76 in Scopus. Of 103 articles, 20 articles were removed for the sake of duplicity. The abstracts of 83 articles were evaluated, and 72 articles were excluded as they did not match the criteria of this study. After eliminating duplicates and titles/abstracts that were not pertinent to the study subject, 11 English-language studies remained. These studies were thoroughly reviewed and evaluated using the following inclusion and exclusion criteria. The following criteria were used to determine their inclusion:

- Only studies that were conducted between 1 January 2010 and 31 December 2020 were reviewed.
- Only articles published in English-written peer-reviewed journals in the mentioned two databases were included.

- Articles reporting the results from experimental/quasi-experimental research were evaluated.
- The primary outcome focused on the use of mobile apps to improve English language teaching.
- Articles conducted within the Bangladeshi context were collected.

The following criteria were used to exclude irrelevant articles:

- Research that was not conducted on the use of the mobile device in ELT, i.e., [41,42].
- Research that was not performed within the Bangladeshi context.
- Studies that were conference papers [21,43], review studies, and book chapters [44] were also excluded.

### 3.3. Data Analysis Process

After selecting the articles, the authors made a database with Mendeley Referencing software. From this database, all the articles are exported as a research information system (.ris) file for importing into NVIVO, the qualitative data analysis software. By importing this ris file into NVIVO, the trends and themes were extracted from the articles to find out the answers to the following research questions: (1) What are the main topics of the detected studies concerning the use of mobile phones in ELT? (2) Is the use of mobile devices effective in ELT in Bangladesh? This systematic literature review analyzed all the articles qualitatively to categorize themes and sub-themes, highlighting the research objectives as a text description and visualization. Two components were considered, one mechanical and one interpretative, the first organizing the data according to the study topics and the second deciding which data were significant in relation to the research questions.

## 4. Results

The empirical studies reviewed for this synthesis ranged from 2010 to 2020. Eleven studies were matched with the criteria of this study that dealt with mobile phones in ELT in Bangladesh. Of these 11 articles, 8 studies were conducted employing a qualitative research design; 5 studies reported on the results of the project, specifically English in Action; and 1 study employed qualitatively the results of six projects held in the Philippines, Mongolia, Thailand, India, and Bangladesh. These projects explored mobile phones for the professional development of secondary and higher secondary teachers for teaching and learning the English language in Bangladesh. Most of the studies conducted (9 out of 11) were based on the qualitative research design. Quantitative and mixed methods were employed, one for each. Most of the studies were conducted targeting the development of language skills of university-level students using mobile phones as a learning tool. Interview and textual documents were the most-used data collection instruments in these studies. A summary of reviewed studies is presented in Table 1 below. The majority of the studies (5 out of 11) were performed between 2010 and 2011. The findings indicate that the use of mobile phones in ELT is still in its infant stage in Bangladesh. The objectives of these studies mainly focus on the effectiveness of mobile phones for facilitating learning environments, teachers' professional development, teachers' experience, and the ways and learners' readiness of using mobile phones in ELT. The findings, however, do not describe what level or to what extent learners improved their language skills by using mobile phones. These studies use mobile phones as tools of instruction with different functions and presentations, i.e., mobile phones, SMS, video recordings, teleconferencing, and smartphones. The research was conducted among university students, teachers of primary and secondary school, and practitioners. This section is divided by subheadings to illustrate the significant findings in the alignment of the research questions.

**Table 1.** A summary of the findings from the selected studies.

| Study | Objective | Respondents | Used Tools | Data Collection Instruments | Findings |
|---|---|---|---|---|---|
| [45] | To enhance the interactivity in distance education using mobile phones | University students (9), teachers (4), and project manager (1). | Mobile phones, SMS, and video recordings | Observations and interviews. | The use of mobile phones increased interactivity with teachers, peers and learning materials for prompting learners' language skills. |
| [46] | To identify the evidence of the role of mobile phones in facilitating m-learning in contributing to improved educational outcomes | Six projects (277, 66, 52,56 & 23) | SMS; MMS; teleconferencing | Textual and numerical documents. | There is an important evidence of mobile phones facilitating increased access. However, much less evidence exists as to how mobiles promote new learning for developing reading, writing, speaking and listening skills. |
| [20] | To investigate teachers' experiences on the use of mobile phones for learning and teaching English | 12 school teachers | Mobile phones | Classroom observation and semi-structure interview. | The use of mobile phones facilitates access to learning, improving the quality of teacher education and training for teaching English in the EFL context, which supports the development of learners' language skills. |
| [47] | To investigate the extent of the use of mobile phones for creating interactive and sustainable learning environments for learning the English language. | Practitioners (11) and researchers (11) | Mobile phones | Artifacts (feedback during lecture, attendance, feedback during the airing of lecture). | Feasibility and usability of mobile phones increase the learners' level of language skills. |
| [48] | To investigate the evidence of an effective and innovative professional development using mobile phones for improving communicative English language teaching. | English in Action project (200 secondary and 400 primary school teachers) | Mobile phones | Classroom observations and teachers' experience. | The use of mobile phones increases the level of communication skills in English through promoting Communicative Language Teaching (CLT.) |
| [49] | To discuss the way the mobile phone has become increasingly relevant to learners in Bangladesh | English in Action project (25 million primary and secondary students) | Mobile phones | Surveys by the EIA, published and unpublished sources, seminar/conference papers, newspaper articles. | Significant improvement of learners' reading and writing skills are achieved. |
| [50] | To investigate the role of mobile phones for Teachers' professional development for enhancing English language teaching | English in Action project (200 secondary and 400 primary school teachers) | Mobile phones | Classroom observations and teachers' experience. | Learners' English language skills are obtained. |
| [51] | To examine the role of mobile phones in the English in Action project for teachers' continuing professional development (CPD) for increasing students' English language competence | English in Action project (4500) | Mobile phones | Content analysis of pilot tests. | Mobile phones support the increase of students' English language competence. |
| [52] | To investigate different ways of using mobile devices for student learning in higher education | 16 University Students | Mobile phones | Semi-structure interviews. | The use of mobile phones provides management of learning materials, effective and collaborative learning for developing learners' language skills. |
| [53] | To explore the effects of variety-seeking (VS) intention from mobile phone usage on students' academic performance (AP). | 322 university Students | Mobile phones | Survey questionnaire. | There is a strong positive relationship between mobile usage and language skills. |
| [54] | To understand EFL learners' readiness for using smartphones for developing oral skills | 61 university students | Smartphones | Survey and interview. | Learners have a high level of readiness for learning oral skills with smartphones. |

### 4.1. Major Research Trends of the Studies

The primary research trends that have emerged from the analysis are the use of the mobile phones for teachers' professional development (TPD) in English language teaching (ELT) [20,33,34], delivery of quality education [29,32], facilitating learning [21,29], increasing interactivity [28], learners' readiness [9], mobile-assisted e-learning framework [26], investigation of learners' concepts [13] and ways of using a mobile device for ELT [13], presented in Figure 2 below. The maximum number of studies dealt with English language teachers' professional development using mobile phones. The research trends on the use of mobile devices for the media of quality education and facilitating learning come second, and the other research trends are investigated one at a time.

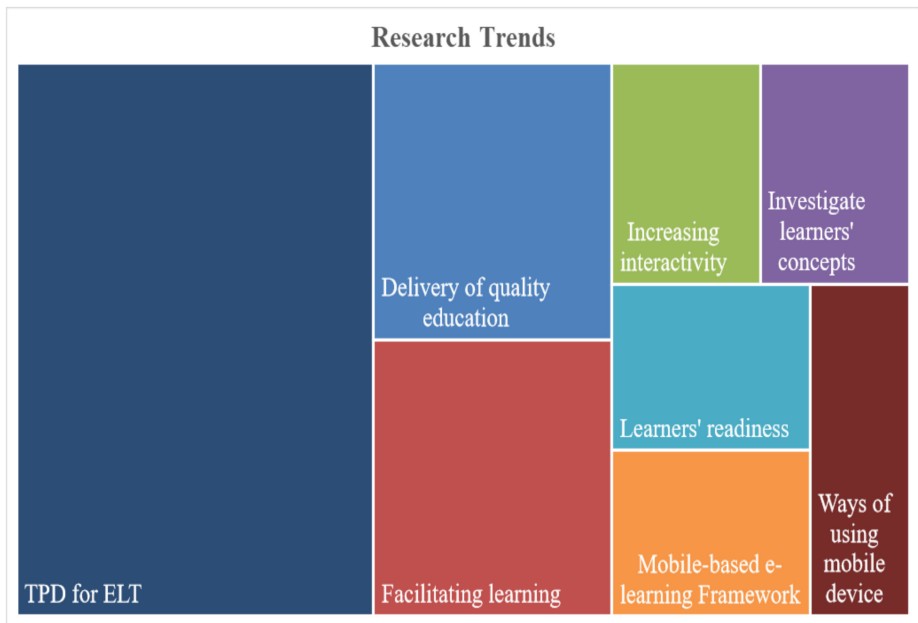

**Figure 2.** Major research trends of the detected studies.

The majority of the studies are held in the beginning stage of commencing mobile devices in ELT in Bangladesh from 2010 to 2011 [20,45–48]. They are aimed at exploring mobile phones for developing primary and secondary English teachers' professional development (TPD) for teaching the English language in terms of making students effective in reading, writing, speaking, and listening skills. Shohel et al. [20] elicited Bangladeshi teachers' experience of taking part in the initial research on an information and communications technology-enhanced supported open distance learning program of professional development in English language teaching. Their findings show that the use of new mobile technologies improves both access to learning and the quality of teacher education and training. The studies that dealt with TPD convey the message that if teachers are efficient using mobile devices in ELT, it will result in more benefits. The results of these studies also confirm that teachers who underwent training on TPD generated many benefits for developing learners' language skills. At this stage, interactivity was achieved by the use of mobile phones, SMS, and video recordings, which allowed students to connect with teachers, other students, and the learning content [45]. Students enjoyed the interactivity. In addition, they found the immediate feedback via a mobile device to be a great motivator [29].

However, the implementation of mobile devices in the initial stage of learning resulted in some challenges, such as organizational, professional, technical, and social [47]. For instance, the study by Grönlund and Islam [47] suggests that the organization should be enhanced by technical support, i.e., SMS server, teacher education, and changes to the course design to avoid organizational challenges. To avoid professional challenges,

teachers must know how to interact with this technology. Removing social challenges means that students should be willing to become independent learners. To enhance and extend the reach of teaching and learning with the use mobile phones, effective and innovative professional development intervention is needed for teachers [47]. Thus, the use of mobile devices should result in innovative professional development among teachers that support improving learners' communicative English language acquisition [20,50].

The majority of studies reported on the project "English in Action," which provides English lessons to students via mobile phones [45,46,48–51]. The project was organized by the government of Bangladesh and subsequently funded (£50 million) by the United Kingdom's Department for International Development (DfID). It aimed to assist 25 million people in Bangladesh to improve their English language skills over 9 years (2008–2017). This project integrated mobile technologies into a school-based teacher professional development program to provide new opportunities for teachers and students to acquire English at levels that enable them to participate more fully in sustainable economic life [53]. This project targeted learners from rural and remote regions to deliver quality education to improve their English language skills [46]. The use of mobiles increased interactivity that supports learners in their English language acquisition [45]. The other studies investigated learners' readiness [54] and the concepts [52] of using mobile devices for learning English language skills. Researchers also targeted different ways of using mobile devices in ELL as a medium for communication, a medium for management of learning materials, a tool for effective learning, a means of collaborative learning, and a means of developing new ideas of using mobiles [52]. Thus, researchers explored the framework of mobile-based language pedagogy for developing learners' English skills [43].

### 4.2. The Effectiveness of Mobile Devices in ELL

The results of the analysis show that mobile devices, respectively the mobile phone, have both a positive and negative effect on learners' learning English language (Figure 3). The themes that emerged from the survey of the previous studies show that the use of mobile devices has more positive (66%) than negative (34%) effects on ELT in Bangladesh. The themes that support the effectiveness of using mobile devices in ELL in Bangladesh are facilitating learning [45,46], feasible learning [46], affordability [54], availability [54], effective as face-face learning [46], new environment for teaching [20], self-supporting for TPD [20], and ubiquitous learning [49].

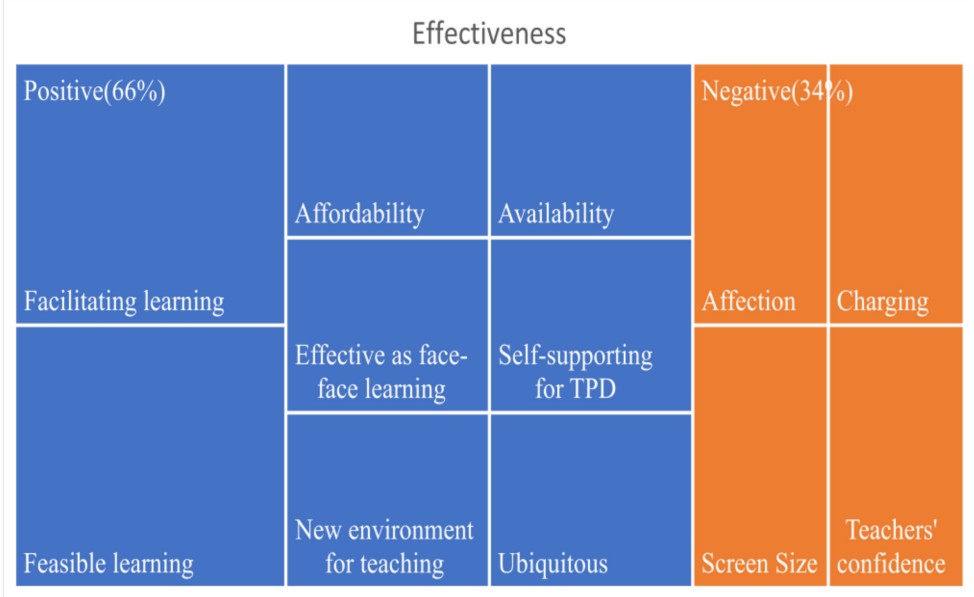

**Figure 3.** Themes on the effectiveness of using mobile phones in ELT.

Among the themes that support the positive effect of the use of mobile phones, facilitating and feasible learning are ranked highest in these studies. The study [45] found that the use of mobile phones allowed interaction with teachers, peers, and educational materials through SMS, which facilitated learning environment. Thus, the use of mobile phones extends access to quality training in a more affordable manner, which facilitates a new learning environment [46]. Moreover, the use of mobile phones for developing learners' language is feasible [47]. The use of mobile phones is effective for language teaching and learning because both teachers and students have mobile phones at their hands. They can use and handle it for their daily communication purposes smoothly and effectively [9]. The findings indicate that the use of mobile devices for learning and teaching language skills is as effective as face-to-face learning [46]. Thus, the use of mobile devices in the language classroom has multiple positive effects on the development of learners' English language skills in Bangladesh.

On the contrary, the results of this review study show that there are some challenges for the effective use of mobile devices in language classes. The themes that challenge this effectiveness are affection [53], charging [51], a lack of teachers' confidence [20], and screen size [42]. Hossain et al. [53] identified that students were affected by mobile phone use. In fact, students tended to continuously use mobile devices for social networking, which had an adverse effect on their academic performance. The other significant problem was that many teachers struggled to charge their mobile device due to the erratic electricity supply in Bangladesh. The findings revealed that teachers placed a higher premium on charging their personal mobile phones while using them in the classroom [51]. The study by Shohel et al. [20] pointed out that teachers lacked confidence because they were using unfamiliar mobile technologies (the iPod and speakers) or introducing the new technologies into classroom practice. Technological challenges, i.e., screen size, was one of the barriers to effective m-learning. This technical issue demonstrates that the quality of the software and hardware is instrumental to the success of m-learning modalities [43].

The results show that the use of mobile devices plays a significant role in developing learners' English language skills in Bangladesh. Furthermore, the results indicate that the mobile device is very effective for creating an innovative and interactive language learning and teaching environment in an EFL context in Bangladesh. Although there are some challenges for the effectiveness of using mobile devices, their application in language learning and teaching is effective.

## 5. Discussion

In response to the first research question, the results show that the main topics/research trends include the use of the mobile phones for teachers' professional development (TPD), the delivery of quality education facilitating learning, increasing interactivity, learners' readiness, mobile-assisted e-learning frameworks, investigation of learners' concepts, as well as ways of using a mobile device for ELT. The maximum studies dealt with mobile phones for teacher professional development to improve learners' language skills.

The findings further reveal that mobile devices are introduced in order to achieve better results in language teaching. This occurred in the project 'English in Action.' The purpose of this project was to enhance technology-based language teaching and learning for the sustainable education and economy of Bangladesh. Thus, the findings of the majority of the detected studies report on the practical experience of teachers and students when using mobile phones in ELT. In addition, the findings describe learners' perception and readiness to use mobile phones for learning language skills. In this regard, both teachers and students reported that they had had a positive attitude to using mobile phones. Apart from that, both teachers and students, i.e., adults use mobile phones for their daily communication needs.

However, research studies with either a short-term or long-term intervention on the use of mobile phones in ELT are rare. In fact, research on the use of mobile phones in ELT in Bangladesh is static. This finding contradicts the findings of the previous studies by Duman et al. [16] and Shadiev et al. [36]. The findings of these review studies indicate that

research in the field of using mobile devices accelerated worldwide after the year of 2008 that reached at maximum in 2017 [16]. The findings of this review also reveal that the most papers were published in 2017, while the fewest were published in 2014 [36]. However, no review studies have been conducted on the research of the use of mobile devices in an EFL context. The evidence from this study clearly illustrates that fewer studies were performed on the use of mobile phones.

The results of this study show a scarcity of empirical studies on the use of mobile phones in ELT in Bangladesh. The maximum number of studies originated in 2010 when the project "English in Action" was in the initial stage of investigating the use of mobile phones in ELT. Except for these projects, most studies investigated university students' perception of using this device without any teaching intervention. Most of the studies employed a qualitative research design using document analysis and interview data. They predominantly focused on how and in what ways mobile devices were used for the benefits of students rather than to what extent students developed their language skills. This finding is in contrast with the findings of Duman et al. [16], who found that the majority of the studies had applied quantitative research methods.

The results in response to the research question on the effectiveness of mobile phones show both positive and negative effects on learning English language skills in Bangladesh. Nevertheless, the use of mobile phones appears to have a more positive effect than a negative one. This device has a primarily positive effect on a facilitating and feasible learning environment for promoting English language learning. As both teachers and students can buy this device, the ubiquitous and new learning environment prevails for learners in learning English language skills. The results show that the use of mobile phones is effective for facilitating and providing feasible learning environments in an EFL context. The findings of this review study are partially aligned with the review study by Klimova and Zamborova [29]. In their study, the findings indicate that mobile learning is becoming a more prominent aspect of education, owing to the fact that it provides an excellent opportunity for foreign language learning. Overall, the research studies illustrate that the effect of mobile phones is mostly found in developing learners' writing, reading and vocabulary [16,41].

In spite of the effectiveness of mobile devices in ELT, there are some challenges to using this device, such as a charging issue, screen size, and teachers' lack of confidence in integrating it into the face-to-face classroom. Despite these shortcomings, mobile devices also foster a social networking tendency among younger learners that negatively effects learners' academic performance. These findings are absent in the other review studies found in the literature.

Although this review study illustrates both positive and negative effects of the use of mobile devices in ELT in Bangladesh, its results suggest that mobile devices have more positive effects than negative ones. In addition, challenges to the effectiveness of the use of mobile phones in ELT can be solved. The authors of this review study believe that the environment for implementing a mobile device in ELT, particularly due to the COVID-19 pandemic, is much better now than any other time.

This review study points out that the use of mobile applications in English language teaching (ELT) is not widely studied in Bangladesh. However, there is potential for developing research on the use of mobile phones in ELT. The findings of this review study indicate that mobile device is very effective for developing learners' English skills, facilitating the effective, feasible and ubiquitous learning process. This study also suggests that both teachers and students are ready and have clear idea of how mobile phones could be integrated into traditional, face-to-face teaching. Nevertheless, the issue of using mobile phones in mainstream classroom activities is still rare. Although the introduction of mobile phones in ELT through teachers' professional development (TPD) was found in 2009, there is still little empirical evidence in the literature. Moreover, there is a lack of evidence about which languages skills students can learn effectively with this device.

Generally, there is potential for developing research on the use of mobile devices in ELT, but it should be randomized and controlled, with larger populations and more extended intervention periods. The research presented here encourages further investigation into mobile devices for a specific language skill, i.e., reading, writing, listening, and speaking. Moreover, there is a need to perform more empirical research in Bangladesh to examine which mobile phone applications are effective for which language skills. Additionally, there is no alternative way to integrate mobile devices into language pedagogy for sustainable and practical education. To fulfill the target of VISION 2041 of the Government of the people of Bangladesh for sustainable education and economy, the integration of the mobile phone into existing teaching methods in ELT classrooms is essential.

## 6. Conclusions

This review study surveyed the empirical research on using mobile phones in ELT published in Scopus and Web of Science indexing journals from 2010 to 2020 in Bangladesh. The findings show that the research topics/trends of the detected studies especially include teachers' professional development using mobile phones for teaching language. Furthermore, other topics concentrate on the issues, such as how the use of mobile phones works as a delivery tool of quality, facilitating learning, increasing interactivity, learners' readiness and teaching techniques of ELT.

In addition, the findings illustrate that the use of mobile phones is effective in ELT through facilitating, feasible, ubiquitous, and effective learning environments with some limitations, i.e., an issue with charging, a small screen, affection, and a lack of teacher confidence. The results of this systematic review confirm that the use of mobile devices is effective for teaching ELT in Bangladesh. Nevertheless, very few studies have been conducted during this time span. In addition, there is a lack of empirical studies with the intention to observe and justify the effect of mobile phones on developing learners' language skills. There is also a lack of evidence describing which mobile applications are effective for developing the relevant language skills.

This review study has some limitations as it only includes (1) the studies published in two reputable databases, namely Scopus and Web of Science, (2) the studies published from 2010 to 2020, (3) the studies conducted in the context of Bangladesh, and (4) the studies conducted following qualitative, quantitative, and mixed-method research designs. All these limitations might affect the overall conclusions of this systematic review.

Despite the shortcomings described above, the results of this systematic review might be applicable in the context of similarly developing countries, as well as triggering empirical research in the field of technology-enhanced ELT in these countries.

**Author Contributions:** Conceptualization, B.K., P.M.I.S.; methodology, B.K., P.M.I.S.; software, N/A; validation, B.K. and P.M.I.S.; formal analysis, B.K.; investigation, P.M.I.S.; resources, P.M.I.S.; data curation, B.K., P.M.I.S.; writing—original draft preparation, P.M.I.S., B.K., H.H.; writing—P.M.I.S., B.K.; visualization, P.M.I.S.; supervision, B.K.; project administration, N/A; funding acquisition, N/A. All authors have read and agreed to the published version of the manuscript.

**Funding:** This research received no external funding.

**Institutional Review Board Statement:** Not applicable.

**Informed Consent Statement:** Not applicable.

**Data Availability Statement:** Not applicable.

**Acknowledgments:** This paper is supported by the project Excellence 2202, run at the Faculty of Informatics and Management of the University of Hradec Kralove, Czech Republic, in 2021.

**Conflicts of Interest:** The authors declare no conflict of interest.

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
