# Peer review of "Use of Mobile Phones in Teaching English in Bangladesh: A Systematic Review (2010–2020)"

_sustainability, doi:10.3390/su13105674_

Round 1
Reviewer 1 Report
The paper is extremely short.
Even if it is a Review article, it is too short and appears incomplete and in need of enhancement and improvement.
If seriously considered for publication, it needs to be expanded and to operate changes in its structure.
Nonetheless, the main question is inherent in the relevance of the article in itself.
Is the article academically significant?
Does the article really add anything to the current debate or to the actual knowledge of its topic?
It is complicate to answer to these questions and the idea is, unfortunately, to have to answer "no".
At the technical level,
- the Introduction is ok, but, in itself, is a sort of preview of the weakeness of the paper as a whole. It could be expanded, explaining better the relevance of the topic in itself and 'why' the topic is approached by the article in the way it is approached;
- the Literature Review is almost inexistent and 'hidden'. Conversely, the Literature Review must have a proper and dedicated section, not connected with Methodology or other aspects - it needs to be considerably enhanced and expanded, and to include most of the currently available (and significant) academic literature, also general works, which would improve the understanding of the readers and trigger further research by the actual audience; moreover, the Literature Review should be written according to a comparative approach, and that would enhance tremendously such an inexplicably absent section. The advice is to implement a dedicated 'window' to the Literature Review, entitled Literature Review and separated from the other sections;
- the Methodology needs a 'chapter' on its own and has to be comprehensively explained, with the main aim of reproducibility and with an attention to the general, non-specialized, audience (the readers);
- Results are ok, possibly the only 'ok' section of the paper, they could also be enhanced a little; nonetheless, if compared to the other parts, they are valid;
- it is inappropriate - and relatively weird - to associate Discussion and Conclusion, and the related section, in the article, is extremely awkward. A proper and dedicated section should include only the Discussion, expanded considerably, both at the level of analysis and hermeneutic comment of the Results, with a direct stress on the significance of the results themselves (if any) and of the article as a whole;
- Conclusion has to be a separate section, 'mirroring' salient elements of the Introduction and corroborating the analysis of the Discussion;
- at the level of expansion in itself, the paper should be increased of at least 5 pages and, also that way, the article could be still considered quite incomplete;
- the English language is ok and also not ok; it is understandable and almost always quite readable, but it would need an additional editorial care (or a revision by a native speaker or near-native speaker), to achieve the level required by an academic publication written in English on a leading international scientific Journal.
Nonetheless, the question is always connected with how relevant the article in itself is and to what extent it adds important contributions to the knowledge of the topic it deals with - the almost total absence of documentation inherent in the Bangladesh context could make the article valuable to fill a gap or, at least, to open a research direction, but the article in itself is too incomplete and negligible, in its current shape, to accomplish a task like that.
Author Response
Dear Reviewer,
We would like to thank you very much for your insightful comments. We have tried hard to implement all of them into the manuscript by which we believe the whole article has been improved significantly. We really appreciate your help with the manuscript and hope you will accept the improvements. Please see the attached file, as well as the revised manuscript about the changes.
Authors

Reviewer 2 Report
- The introduction presents a good background for the study but could be improved with a more clearly articulated problem statement and research questions
- This research needs deep literature on the problem statement and research questions. So, related literature should be in its own section and extended to bring out the limitations of the present research, leading to how the proposed research model-study overcomes these. There is more recent literature that could be included, and it was noted that there were no recent references.
- Furthermore, the reference list of new publications is a little bit weak. There are not enough studies from European or western countries. Before I can make a final decision on the paper, please refer to more references. It is suggested that the author(s) can consider the following papers related to the use of new forms of technology, use of mobile devices education etc. to strengthen the background and conclusions of the study:
- Papadakis, S.; Vaiopoulou, J.; Kalogiannakis, M.; Stamovlasis, D. Developing and Exploring an Evaluation Tool for Educational Apps (E.T.E.A.) Targeting Kindergarten Children. Sustainability 2020, 12, 4201.
- Papadakis, S., Kalogiannakis, M., Sifaki, E., & Vidakis, N. (2018). Evaluating Moodle use via Smart Mobile Phones. A case study in a Greek University. EAI Endorsed Transactions on Creative Technologies, 5(16).
- Papadakis, S., & Kalogiannakis, M. (2020). A Research Synthesis of the Real Value of Self-Proclaimed Mobile Educational Applications for Young Children. In S. Papadakis, & M. Kalogiannakis (Eds.), Mobile Learning Applications in Early Childhood Education (pp. 1-19). Hershey, PA: IGI Global. doi: 10.4018/978-1-7998-1486-3.ch001
- The paper needs to explain more clearly and in enough depth the research approach of the study.
- The methods should be adequately described, and show how the methods used will provide the answers to the questions.
- The sampling technique and the data gathering instruments need to be described in sufficient detail, e.g., explain how the questionnaire was developed. Similarly, it needs to be made clear how the data analysis methods used in the paper are appropriate for the analysis of the data obtained
- Discussion of results needs to be clearer, how do findings relate to curriculum development models?
- Conclusion should reiterate the purpose of the paper and state how the study has answered the research questions.
* In preparing a revised manuscript, please also include a table of how you have responded to each of the issues listed above point by point.
Author Response

(The authors gave the same response as above.)

Round 2
Reviewer 1 Report
A lot of work has been done.
At least, the Authors have followed the Reviewer advice and the article appears as a better piece of research, now, especially more comprehensive.
At the level of implementation of the different parts and their contents, now the article seems better and robust.
What still remains is the question about the relevance of the article in itself, but, at the level of structure, the work can be accepted.
A word of true appreciation to the Authors for the quite good revision.
Reviewer 2 Report
I am generally very sympathetic towards the project of this paper.
I applaud all the efforts of the author(s) for the revised version of this manuscript.
Overall, the literature review is considered adequate.
The study is well designed and executed.